# Efficacy of Molnupiravir in Reducing the Risk of Severe Outcomes in Patients with SARS-CoV-2 Infection: A Real-Life Full-Matched Case–Control Study (SAVALO Study)

**DOI:** 10.3390/microorganisms13030669

**Published:** 2025-03-15

**Authors:** Ivan Gentile, Riccardo Scotto, Maria Michela Scirocco, Francesco Di Brizzi, Federica Cuccurullo, Maria Silvitelli, Luigi Ametrano, Francesco Antimo Alfè, Daria Pietroluongo, Irene Irace, Mariarosaria Chiariello, Noemi De Felice, Simone Severino, Giulio Viceconte, Nicola Schiano Moriello, Alberto Enrico Maraolo, Antonio Riccardo Buonomo, Agnese Giaccone

**Affiliations:** 1Department of Clinical Medicine and Surgery—Section of Infectious Diseases, University of Naples Federico II, 80131 Naples, Italy; ivan.gentile@unina.it (I.G.); mmichelascirocco@hotmail.com (M.M.S.); francescodibrizzi@gmail.com (F.D.B.); federicacuccurullo94@gmail.com (F.C.); mariasilvitelli94@gmail.com (M.S.); luigi.ametrano@outlook.com (L.A.); francescoantimoalfe@gmail.com (F.A.A.); pietroluongo.daria@gmail.com (D.P.); irisirene@hotmail.it (I.I.); chiariello.mr@outlook.it (M.C.); noemidefelice@gmail.com (N.D.F.); simoneseverino5f@gmail.com (S.S.); giulio.viceconte@gmail.com (G.V.); veghan@gmail.com (N.S.M.); albertomaraolo@yahoo.com (A.E.M.); antonioriccardobuonomo@gmail.com (A.R.B.); 2Department of Infectious Diseases, Unit of Geriatric Infectious Diseases, AORN Ospedali dei Colli, Cotugno Hospital, 80131 Naples, Italy; agnesegiaccone94@gmail.com

**Keywords:** SARS-CoV-2, COVID-19, molnupiravir, early treatment

## Abstract

We conducted a real-life case–control study among outpatients with Omicron SARS-CoV-2 infection to assess the effectiveness of molnupiravir (MNP) in reducing hospital admission, admission to the intensive care unit, and death at day 28. Cases were SARS-CoV-2-positive patients seeking medical care within five days of symptom onset from 1 January to 31 December 2022, who received MNP. Controls were selected from a regional database among positive subjects who did not receive antiviral treatment for SARS-CoV-2. A total of 1382 patients were included (146 cases, 1236 controls). Vaccinated patients had a lower risk of mortality and of the composite outcome (hospital admission, ICU admission, or all-cause death) than unvaccinated ones (0.6% vs. 7.8%, *p* < 0.001 and 2% vs. 7.8%, *p* = 0.001, respectively). After full-matching propensity score analysis, MNP-treated subjects had a lower incidence of the composite outcome, although no effect was observed on individual outcomes. In subgroup analyses by vaccination status, MNP was effective in preventing all outcomes among unvaccinated patients and reduced the risk of ICU admission in both vaccinated and unvaccinated patients. Molnupiravir treatment effectively reduced the composite outcome risk in outpatients with SARS-CoV-2 infection, with a more pronounced benefit in unvaccinated patients. These findings highlight MNP’s potential to help prevent disease progression in high-risk patients, thereby supporting its role as an outpatient therapeutic option for COVID-19.

## 1. Introduction

Since its emergence in late 2019, SARS-CoV-2 has continued to be a major cause of morbidity and mortality worldwide [1]. Although there are limited options for outpatient treatment, the introduction of effective oral antiviral therapies in the last two years has been a turning point for the early treatment of COVID-19 in ambulatory care settings to prevent deterioration in patients at risk of progression [2,3]. Molnupiravir (MNP) is an orally administered ribonucleoside prodrug of N-hydroxycytidine that is metabolized in vivo to the ribonucleoside analog β-D-N4-hydroxycytidine. It exerts its antiviral effect by causing errors in the SARS-CoV-2 RNA genome during replication, ultimately leading to error catastrophe and inhibiting viral propagation [4,5,6]. Compared to other “-ravir” drugs such as nirmatrelvir/ritonavir, MNP has fewer significant drug–drug interactions, making it more suitable for individuals on multiple concomitant medications [7]. This feature may be especially relevant in outpatient settings, where a streamlined treatment approach is needed to promptly address early COVID-19 in vulnerable or high-risk patients who cannot receive nirmatrelvir/ritonavir due to contraindications or complex pharmacological profiles. Supported by data from the phase 3, double-blind, randomized, placebo-controlled MOVe-OUT trial, MNP received emergency use authorization by the U.S. Food and Drug Administration in December 2021 for the treatment of non-hospitalized patients with symptomatic COVID-19 at high risk of progression to severe COVID-19 [4,8]. However, the MOVe-OUT trial was conducted among unvaccinated patients before the emergence of the Omicron (B.1.1.529) variant and subsequent sub-lineages. Therefore, collecting real-life data on the effectiveness of oral antiviral agents in a mostly immunized population with predominant Omicron variant infections has become a research priority. Thus, the present case–control study aims to assess the efficacy of MNP in preventing hospital admission, admission to the intensive care unit (ICU), and death in a real-life population, mostly vaccinated or possessing hybrid immunity, with SARS-CoV-2 Omicron variants infection.

## 2. Materials and Methods

### 2.1. Study Design and Population

This study was a retrospective case–control analysis. Inclusion criteria were as follows: (1) a positive SARS-CoV-2 test between January 2022 and December 2022; (2) aged 18 years or older; and (3) no prior treatment with COVID-19 antiviral therapies or monoclonal antibodies, either alone or in combination with MNP. Exclusion criteria included the following: (1) individuals under the age of 18; (2) patients who had already received any other antiviral therapy or monoclonal antibodies for SARS-CoV-2 prior to study enrollment; and (3) patients seeking medical care after day 5 of symptom onset (i.e., asymptomatic or presenting beyond the recommended time window for early treatment).

Cases in this study were defined as individuals who sought outpatient care at the Infectious Diseases Unit of Federico II University in Naples, Italy, and were treated with MNP within five days of symptom onset. All patients who received treatment had risk factors for developing severe COVID-19, as outlined by the Italian Medicines Agency (AIFA) [9].

Control participants were selected from a regional database of individuals who tested positive for SARS-CoV-2 during the study period. Controls were identified and validated through telephone interviews designed by the authors and conducted by healthcare professionals (see Appendix A for the full interview script).

This study was conducted in line with the principles of the Declaration of Helsinki and received approval from the Ethics Committee ‘Comitato Etico Università Federico II–A.O.R.N. A. Cardarelli’ (protocol number 0015191, 22^nd^ March 2023).

### 2.2. Data Collection and Definitions

We collected comprehensive data on patients’ demographic characteristics, comorbid conditions, vaccination status, and the predominant SARS-CoV-2 variant in our region during each phase of the study period. Specifically, we recorded each participant’s age, sex, and, when available, body mass index (BMI). We also noted all relevant comorbidities (e.g., diabetes, hypertension, chronic heart disease, COPD, chronic kidney disease, liver disease, neurological disorders) and whether the participant was immunodeficient (e.g., solid organ transplant recipient, active malignancy, or on immunosuppressive therapy). Vaccination data included the total number of SARS-CoV-2 vaccine doses received, as well as the interval from the last dose to the first positive test when available. Additionally, we documented the prevalent Omicron subvariant in circulation (BA.1, BA.2, BA.5, etc.) at the time of each patient’s infection. Patients were classified as fully vaccinated if they had received at least two doses of a COVID-19 vaccine. We also used the Monoclonal Antibody Screening Score (MASS), a validated clinical tool for identifying patients at high risk for severe COVID-19 who may benefit from early outpatient therapies. Specifically, the MASS incorporates points based on age, body mass index, comorbidities (e.g., diabetes, chronic heart disease, respiratory disease), and immunosuppression status. A higher MASS indicates an increased likelihood of disease progression, providing an objective measure to guide treatment decisions [10]. For each patient, we calculated the MASS and we also formulated a simplified comorbidity score: patients with three or more comorbidities were assigned 2 points, those with one or two comorbidities received 1 point, and those without any comorbidities were assigned 0 points.

### 2.3. Propensity Score Matching

We performed a propensity score (PS)-matched analysis to mitigate selection bias and ensure an equitable distribution of confounding variables between cases and controls [11]. Our approach to PS methods followed the guidelines recommended by Eikenboom et al., aimed at enhancing the quality of research on the effectiveness of antimicrobial therapies [12]. In detail, the model was developed for each subject, assigning a probability of receiving molnupiravir based on baseline characteristics such as age, sex, vaccination status, comorbid conditions, MASS, simplified comorbidity score, and the predominant SARS-CoV-2 variant. Probabilities were calculated using probit regression. The balance of PS matching was evaluated by measuring the absolute standardized mean difference (SMD) of covariates, both continuous and categorical, between the groups. A value below 0.1 was deemed to indicate an acceptable balance.

We assessed various matching algorithms, including greedy matching (with different ratios and calipers), optimal matching, and full matching [13]. The latter yielded the optimal balance while preserving the entire sample size. This approach allocated each unit in the sample to a subclass, with subclasses containing either one treated unit and one or more control units, or one control unit and one or more treated units. The number of subclasses and the allocation of units were optimized to minimize the sum of absolute within-subclass distances in the matched sample [14].

The chosen estimand was the average treatment effect on the treated (ATT), which represents the average effect of treatment among those who received it. This estimand addresses the following fundamental question: “should medical providers refrain from withholding treatment from those currently receiving it?” [15]. We estimated marginal effects, comparing the expected potential outcomes under treatment to those under control, and expressed these as odds ratios (ORs) with 95% confidence intervals (CIs). Estimation was conducted using g-computation with a cluster-robust standard error to account for pair membership, focusing on binary outcomes and incorporating covariates via logistic regression [16]. Including covariates in the outcome model after matching serves several purposes: it can enhance precision in the effect estimate, reduce bias due to residual imbalance, and provide a “doubly robust” effect estimate. This means the estimate remains consistent if either the matching sufficiently reduces covariate imbalance or if the outcome model is correctly specified [17].

A secondary moderation analysis was performed to assess whether the treatment effect differed across varying levels of another variable, specifically vaccination status. This was achieved by carrying out matching on the full dataset to ensure balance within each subgroup of the moderating variable [18].

### 2.4. Outcomes

The primary outcomes of the study were the proportions of hospital admissions, intensive care unit (ICU) admissions, and 28-day all-cause mortality among both the treated individuals and the propensity-matched untreated individuals. Additionally, we evaluated a composite outcome, defined as the occurrence of at least one of the three aforementioned events (hospital admission, ICU admission, or all-cause death). We selected these specific endpoints to capture clinically meaningful progression to severe disease in a single measure, a strategy also adopted in previous COVID-19 therapeutic trials [3,4]. The follow-up period extended to 28 days from the onset of symptoms or the first positive SARS-CoV-2 test.

### 2.5. Statistical Analysis

Categorical variables are presented as counts and percentages, while continuous variables are reported as medians with interquartile ranges (IQRs). Pearson’s χ^2^ test was employed to assess differences between groups for categorical variables, and the Mann–Whitney U test was used for continuous variables. A two-sided *p*-value < 0.05 was considered statistically significant for all comparisons.

After matching, the effect of MNP on the outcomes of interest was estimated. All analyses were performed using R, version 4.1.0 (R Core Team), with the following packages: MatchIt, cobalt, and marginal effects.

## 3. Results

The study included a total of 1382 patients. This comprised 146 cases (10.6%) and 1236 controls (89.4%) (Table 1).

Patients treated with molnupiravir exhibited a more complex clinical profile. Specifically, they were older (*p* < 0.001), and they more frequently had hypertension, chronic heart disease, chronic kidney disease (*p* < 0.001), COPD (*p* = 0.024), and immunodeficiency (*p* < 0.001). These patients also exhibited a higher comorbidity score (*p* < 0.001) and a higher MASS (*p* < 0.001), thus reflecting a higher baseline risk of progression to severe COVID-19 in the treated cohort.

Twenty-three patients (1.7%) required hospitalization, and one patient (0.1%) was admitted to ICU. Only 1 out of 146 (0.7%) patients treated with MNP died, while 15 (1.2%) patients among the controls died. No significant differences were observed in any of the outcomes under evaluation between the treated and untreated groups (Table 2).

However, after a full-matching propensity score analysis, MNP was shown to be effective in reducing the rate of the composite outcome, while no effect on hospital admission or death alone was observed (Table 3).

When comparing outcomes after stratification for vaccination status, we observed that patients who received the SARS-CoV-2 vaccine had a lower death rate compared to unvaccinated patients (*p* < 0.001) and a significantly reduced risk of the composite outcome (*p* = 0.001) (Table 4).

When stratifying treated and untreated patients according to their vaccination status, none of the evaluated outcomes showed a significative reduction in the MNP arm (Appendix A). In the subgroup analysis conducted among both cases and controls who received the SARS-CoV-2 vaccine, MNP was shown to reduce the risk of ICU admission both in vaccinated and unvaccinated subjects after full-matched propensity score analysis (*p* < 0.001) (Table 5).

## 4. Discussion

In this retrospective, PS-matched cohort study, we evaluated the real-life effectiveness of MNP in preventing hospitalization, ICU admission, and death among outpatients with Omicron SARS-CoV-2 infection in a highly vaccinated population. We used MASS to systematically quantify each patient’s baseline risk, which proved particularly important given that MNP-treated patients were significantly older and had a higher comorbidity burden. The higher MASS observed in the MNP group underscores that these individuals were indeed more vulnerable to severe outcomes. Nonetheless, the full-matched propensity score provided a fair and balanced comparison between cases and controls (Appendix A; Appendix A). We collected data on the outcomes of MNP-treated patients and untreated patients during the same period to achieve a relative homogeneity in the infecting variants of SARS-CoV-2 between the two groups. Specifically, Omicron BA.1, BA.2, and BA.5 were the most prevalent variants: among cases, we observed a polarization around BA.2 (87.7%), while the variants were more evenly distributed among controls with a prevalence of BA.5 (50.7%). Most of both cases and controls in our study (>90%) had received at least two doses of SARS-CoV-2 vaccine. Given the high rate of immunization of our cohort and the prevalence of Omicron variants, whose virulence is well known to be inferior to previous SARS-CoV-2 variants, a low rate of disease progression and death was expected, in line with worldwide epidemiological data [19]. Notably, 15 deaths were reported among the controls, while only 1 death was recorded in the MNP-receiving group. This is a remarkable finding considering that the latter group included patients with a frailer clinical profile, significantly older mean age, and higher comorbidity scores. A more favorable disease course was confirmed after PS analysis, with a lower incidence of the composite outcome in the MNP-treated group (*p* = 0.013); however, no significant reduction in the separate rate of hospital admission or death was observed. Our results are partially consistent with those of other cohorts. Actually, the large, open-label, randomized controlled PANORAMIC trial assessed the efficacy of early treatment with MNP versus placebo among a highly vaccinated population (94% of participants had received at least three doses of a SARS-CoV-2 vaccine). The trial demonstrated earlier symptom alleviation and faster viral load decline in the MNP-receiving group but no significant reduction in hospital admission and mortality rate (which was around 1% in both treatment arms) [20]. Moreover, subsequent data from the PANORAMIC cohort have been recently published, showing improvements in long-term clinical outcomes among vaccinated patients with SARS-CoV-2 treated with MNP [21]. We reported a similar result in our study, with no significant difference in hospitalization and mortality rate, although we found MNP to be effective in reducing the occurrence of composite outcomes in our highly vaccinated cohort. Moreover, Wong et al., in a retrospective, propensity-score matched analysis on a cohort of hospitalized patients with COVID-19 not requiring oxygen therapy on admission, found that early administration of MNP significantly reduces the risk of all-cause mortality and disease progression (hazard ratio, HR 0.60, 95% CI 0.52–0.69) [3]. Similarly, a propensity score-matched retrospective study conducted among U.S. veterans showed a lower combined 30-day risk of hospitalization or death in MNP-treated participants aged ≥ 65 years versus no treatment (relative risk, RR 0.67, 95% CI 0.46–0.99) [22]. Conversely, some studies reported no significant effect of MNP, as shown by Yip et al. in their retrospective propensity-score matched analysis [23]. Accordingly, Najjar-Debbiny et al. reported a nonsignificant reduced risk of the composite outcome (HR 0.83, 95% CI 0.57–1.21) in the MNP-treated group. However, subgroup analyses showed that MNP was associated with a significant decrease in the risk of the composite outcome in older patients, females, and patients with inadequate COVID-19 vaccination [24]. A recent systematic review of real-world studies reported that MNP was effective in reducing the risk of severe COVID-19 outcomes, particularly among older age groups. The authors reported as potential bias a difference in the baseline characteristics of the MNP-treated group versus the untreated group, with the former including generally older participants and those with more comorbidities than controls. Therefore, they inferred that the actual effectiveness of MNP may have been underestimated [25]. We might draw the same inference for our cohort, where the cases were consistently older and more vulnerable than controls, even after minimizing the differences in the baseline features between the two groups by PS matching. Additionally, all the studies included in the above-mentioned systematic review were conducted when the Omicron subvariants were predominant and a relevant part of the population already had prior immunity to SARS-CoV-2 [25]. It is worth mentioning that, although the MOVe-OUT trial only included unvaccinated patients, in the subgroup analysis of patients with evidence of previous SARS-CoV-2 infection, MNP showed no better outcome compared to placebo [4]. Unvaccinated individuals are indeed more likely to have a higher viral load and a more prolonged viral replication during the initial phase of infection. Molnupiravir works by introducing errors into the viral RNA during replication; thus, it may be more effective in reducing the viral burden and preventing disease progression, especially in the early stages of infection. Moreover, without prior exposure to the virus or vaccination, the immune system in unvaccinated individuals is often less primed to mount an effective defense against the virus, while vaccinated individuals typically have a more rapid and robust immune response that may control the viral replication more effectively, making antiviral therapy less critical for them. Given the high rate of vaccination against COVID-19 among the general population, real-world studies play a crucial role in evaluating the real advantage of MNP employment in patients with prior SARS-CoV-2 immunity (either acquired through vaccination or past infection).

In our study, we observed a significantly higher incidence of death (7.8% vs. 0.6%, *p* < 0.001) and composite outcome (7.8% vs. 2%, *p* = 0.001) among unvaccinated patients, than in vaccinated patients. This finding supports the evidence that vaccination against SARS-CoV-2 is highly effective in preventing clinical deterioration and death regardless of the immune escape of new variants and their limited virulence compared to the wild-type virus. As expected, the most profound effect was observed among unvaccinated individuals, for whom MNP proved effective in reducing the occurrence of all evaluated outcomes, with a remarkably higher benefit from the antiviral therapy compared to fully vaccinated patients. Among the latter, MNP reduced the risk of the composite outcome and hospitalization, but not in a statistically significant fashion. Nonetheless, MNP retained significant efficacy in preventing ICU admission among fully vaccinated subjects. This underscores a potential benefit of MNP regardless of vaccination status. As is the case with many medical interventions, there is likely to be a gradient of benefit for treatment with MNP, with the greatest impact shown in the subjects at the highest risk for progression and without prior immunization [26].

The apparent lack of effectiveness reported by some studies might partly be related to its availability in earlier days when prescribing criteria were more stringent, and MNP was preferentially assigned to more frail and polymedicated patients than those who received nirmatrelvir/ritonavir, perhaps because of the multiple drug–drug interactions associated with the latter [23].

Essentially, our results, along with the recent real-life literature, suggest that MNP may be effective in reducing the risk of unfavorable outcomes in patients with SARS-CoV-2 infection, particularly in higher-risk or unvaccinated patients. However, some studies have reported less pronounced benefits or no significant effect of MNP. For example, Yip et al. found no significant advantage in certain populations, potentially due to differences in the baseline risk profiles, rates of vaccination, timing of antiviral administration, or variations in Omicron sublineages [23]. Similarly, Najjar-Debbiny et al. reported a nonsignificant reduction in the risk of a composite outcome, though MNP appeared more beneficial in older patients or those with incomplete COVID-19 vaccination [24]. Such discrepancies across studies may also stem from local treatment protocols, patient demographics, and the specific definitions of high-risk individuals. Nonetheless, our findings highlight the potential protective role of MNP when targeted appropriately and adjusted for confounding via propensity-score methods. Unfortunately, after the completion of our study, the Scientific Technical Commission of AIFA decided to suspend the use of MNP in March 2023, following the negative opinion issued by the European Medicines Agency (EMA) [27,28]. This decision leaves European patients who need oral antivirals but cannot take nirmatrelvir/ritonavir (e.g., due to insurmountable drug-to-drug interactions) with no other viable therapeutic options apart from a short course of remdesivir, which, however, incurs hospitalization costs. According to the CDC, MNP remains a second-line treatment option for patients with SARS-CoV-2 infection at risk for severe disease who cannot be treated with nirmatrelvir/ritonavir or remdesivir, and MNP continues to be administered in the United States [29]. Additionally, the WHO’s living guidelines on COVID-19 conditionally recommend the use of MNP for patients with non-severe COVID-19 at the highest risk of hospitalization (excluding pregnant and breastfeeding women, and children) [30].

The major strengths of our study are the large number of controls and the use of PS full matching, which allows us to estimate a treatment effect that is ideally free of confounding due to the measured covariates. Indeed, this algorithm contributes to minimizing the confounding effect of a marked disproportion between the two groups in a key variable at baseline, namely the immunodeficiency status that is present in 48.6% vs. 6.5% among cases and controls, respectively. Especially regarding the impact of MNP on vaccinated and unvaccinated patients, the PS analysis enabled us to detect a substantial improvement in all the outcomes that had not previously emerged in the baseline statistical analysis. This approach helped distribute key covariates (e.g., age, comorbidity load, immunodeficiency status) more evenly across both arms, but it cannot completely eliminate residual or unmeasured confounding. Consequently, while our findings highlight the protective role of MNP, particularly among high-risk patients, these results must be interpreted with caution.

We recognize that our study has other limitations, mainly related to its retrospective design. In fact, while data for cases were collected prospectively, data for controls were retrieved retrospectively. Since controls were identified through telephone interviews, some may not have accurately reported baseline features (e.g., vaccination status) or outcomes. For instance, hospital admissions beyond the predefined follow-up window might have been reported as having occurred earlier due to recall bias. Despite efforts to track relatives of deceased patients, it is possible that a significant percentage of controls (or their relatives) who had an unfavorable outcome declined to be interviewed. To mitigate this bias, we employed a standardized interview script and trained healthcare professionals to conduct the telephone interviews, ensuring consistency in data collection. Furthermore, the large size of our control group helps to reduce the proportional impact of any misclassification arising from imperfect recall. Nonetheless, the possibility of under- or overestimation of clinical events (e.g., hospitalization, ICU admission) cannot be fully ruled out. Additionally, the 28-day follow-up period we employed captures only the acute-phase outcomes and may not fully reflect longer-term benefits or late complications potentially associated with MNP. We recognize that extending the follow-up to 60 or 90 days could provide important insights into post-COVID sequelae and other long-term endpoints. In this regard, we are currently conducting a separate study with an extended follow-up to further investigate the persistence of any benefits, as well as any late adverse outcomes, in patients who received molnupiravir. Moreover, our cohort included a relatively homogeneous population, which could limit the extrapolation of these results to more diverse groups, especially in regions with varying genetic backgrounds that may affect drug metabolism (pharmacogenetics). Nonetheless, some real-life studies conducted among different populations yielded similar results, such as those by Wong et al. [3] and Bajema et al. [22], suggesting that different genetic subsets should not undermine the efficacy of MNP. Indeed, the differences observed in cohorts in Hong Kong (Yip et al.) [23] and Israel (Najjar-Debbiny et al.) [24] may partially reflect variation in sublineages, vaccination coverage, or population-specific risk profiles. Future real-world studies in broader and more heterogeneous settings will be essential to clarify whether genetic and environmental factors alter MNP’s effectiveness and thus further refine its global applicability. Finally, a further notable limitation is the imbalance in sample size between the MNP-treated group (146 cases) and the control group (1236 controls). While PSM helped create a balanced comparison by adjusting for key confounders, the smaller sample size of the treated group may have reduced the statistical power to detect significant differences in individual outcomes, such as hospitalization or mortality. This may partly explain why the observed benefit of MNP was more evident for the composite outcome rather than for each endpoint separately. Nevertheless, this sample size imbalance reflects real-world treatment allocation, where MNP was reserved for a subset of high-risk patients based on clinical criteria, rather than being randomly assigned.

## 5. Conclusions

In conclusion, our real-life study demonstrated that, after full-matching the propensity score, molnupiravir can reduce the risk of composite outcome (at least one among hospitalization, ICU admission, and all-cause death) at day 28, irrespective of vaccination status, in highly vaccinated outpatients infected with SARS-CoV-2 Omicron variants.

The beneficial effect of MNP treatment in reducing progression is more pronounced in unvaccinated patients. Its real effectiveness, however, might have been underestimated due to residual unmeasured confounding that was not adjusted by PS.

## Figures and Tables

**Table 1 microorganisms-13-00669-t001:** Baseline features of pre-matched sample (N = 1382).

	Molnupiravir	No Therapy	*p*-Value
n	146	1236	
Male Sex (n, %)	77 (52.7)	551 (44.6)	0.074
Age (median [IQR])	70.00 [59.00, 79.00]	57.00 [46.75, 68.00]	<0.001
Vaccination (%)	138 (94.5)	1142 (92.4)	0.446
Diabetes (%)	13 (8.9)	115 (9.3)	0.995
Hypertension (%)	52 (35.6)	245 (20.6)	<0.001
Chronic heart disease (%)	60 (41.1)	200 (16.2)	<0.001
COPD (%)	18 (12.3)	84 (6.8)	0.024
CKD (%)	11 (7.5)	24 (1.9)	<0.001
Obesity (%)	23 (15.8)	145 (11.7)	0.203
Liver disease (%)	4 (2.7)	14 (1.1)	0.217
Neurological disease (%)	6 (4.1)	46 (3.7)	0.998
Immunodeficiency (%)	71 (48.6)	80 (6.5)	<0.001
Comorbidity score (median [IQR])	1.00 [1.00, 1.00]	0.00 [0.00, 1.00]	<0.001
MASS (median [IQR])	5.00 [3.00, 6.75]	1.00 [0.00, 3.00]	<0.001
Predominant variant (%)			<0.001
- Omicron (not specified)	0 (0.0)	123 (10.0)	
- Omicron_BA.1	18 (12.3)	212 (17.2)	
- Omicron_BA.2	128 (87.7)	271 (21.9)	
- Omicron_BA.5	0 (0.0)	627 (50.7)	
- Omicron_BQ.1	0 (0.0)	3 (0.2)	

IQR: interquartile range; COPD: chronic obstructive pulmonary disease; CKD: chronic kidney disease.

**Table 2 microorganisms-13-00669-t002:** Outcome rates among cases and controls.

	Molnupiravir	No Therapy	*p*-Value
n	146	1236	
Hospital admission (%)	3 (2.1)	20 (1.6)	0.962
ICU admission (%)	0 (0.0)	1 (0.1)	1.000
Death (%)	1 (0.7)	15 (1.2)	0.876
Composite outcome (%)	3 (2.1)	31 (2.5)	0.959

**Table 3 microorganisms-13-00669-t003:** Effect of MNP on different outcomes after full-matched propensity score.

	Point Estimate (OR)	Standard Error	95% Confidence Interval	*p*-Value
Composite outcome	0.353	6.2	0.155–0.805	0.013
Hospital admission	0.744	0.4	0.132–4.18	0.737
ICU admission	*	*	*	*
Death	0.958	0.3	0.701–1.31	0.787

* Algorithm did not converge (too few events).

**Table 4 microorganisms-13-00669-t004:** Outcome rates among included patients according to vaccination status.

	Vaccination	No Vaccination	*p*-Value
n	1280	102	
Hospital admission (%)	21 (1.6)	2 (2.0)	1.000
ICU admission (%)	1 (0.1)	0 (0.0)	1.000
Death (%)	8 (0.6)	8 (7.8)	<0.001
Composite outcome (%)	26 (2.0)	8 (7.8)	0.001

**Table 5 microorganisms-13-00669-t005:** Effect of MNP on different outcomes after full-matched propensity score, according to vaccination status.

	Vaccination	Point Estimate (OR)	Standard Error	95% Confidence Interval	*p*-Value	*p* Subgroup
Composite outcome	Yes	0.596	1.0	0.126–2.82	0.514	<0.001
	No	1.67 × 10^−7^	80.6	8.63 × 10^−9^–3.23 × 10^−6^	<0.001	
Hospital admission	Yes	0.609	0.9	0.127–2.93	0.536	<0.001
	No	2.29 × 10^−5^	84.5	3.17 × 10^−6^–1.66 × 10^−4^	<0.001	
ICU admission	Yes	1.60 × 10^−7^	175.8	2.19 × 10^−8^–1.17 × 10^−6^	<0.001	<0.001
	No	1.75	*	1.75–1.75	<0.001	
Death	Yes	9.46	3.2	0.621–1.44 × 10^2^	0.106	<0.001
	No	2.26 × 10^−8^	101.7	1.17 × 10^−9^–4.38 × 10^−7^	<0.001	

* Algorithm did not converge (too few events).

## Data Availability

The datasets generated and/or analyzed during the current study, along with the study’s Case Report Forms (CRFs), the written informed consent from case participants, and the records of the telephonic interviews conducted with control participants, including their verbal informed consent, are stored by the study’s data manager at the Department of Clinical Medicine and Surgery, University of Naples Federico II, Via Sergio Pansini 5, 80131, Naples, Italy. These can be retrieved and reviewed upon request by any relevant authority.

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
