# Peer review of "Efficacy of Molnupiravir in Reducing the Risk of Severe Outcomes in Patients with SARS-CoV-2 Infection: A Real-Life Full-Matched Case–Control Study (SAVALO Study)"

_microorganisms, 2025, doi:10.3390/microorganisms13030669_

Round 1
Reviewer 1 Report
Comments and Suggestions for Authors
This study investigates the real-world effectiveness of Molnupiravir (MNP) in reducing severe outcomes (hospital admission, ICU admission, and death) in patients with SARS-CoV-2 infection. A full-matched case-control design strengthens the analysis, allowing for a more robust assessment of MNP's impact across vaccinated and unvaccinated individuals. The findings suggest that MNP is beneficial, particularly in unvaccinated patients, and contributes to the ongoing evaluation of antiviral therapies for COVID-19 management. While the manuscript is well-structured and presents valuable data, certain methodological aspects and discussions could be further elaborated to enhance clarity and scientific depth. Below are specific points for improvement.
Minor points
- The study includes a relatively small number of MNP-treated patients (146 cases) compared to the control group (1236 controls). While propensity score matching helps mitigate confounding, the imbalance in sample size may still introduce bias. The authors should explicitly acknowledge this limitation and discuss potential implications for the study's statistical power.
- The study follows patients for 28 days from symptom onset or the first positive test. The authors should justify why this specific timeframe was chosen and discuss whether extending the follow-up period (e.g., to 60 or 90 days) could provide additional insights into the long-term benefits or potential late complications associated with MNP treatment.
- Given that the study population appears relatively homogeneous, there may be limitations in extrapolating these findings to more diverse populations, particularly in regions with different genetic backgrounds that may influence drug metabolism (pharmacogenetics). A discussion on how population heterogeneity could affect the reproducibility of these results would strengthen the manuscript. The authors could compare their findings with other real-world studies conducted in different populations if possible.
Discussion: While the study demonstrates that MNP is particularly effective in unvaccinated individuals, it would be beneficial to expand on the potential immunological and virological mechanisms behind this observation. A brief discussion could enhance the mechanistic understanding of the drug's role in different patient subgroups. The authors could briefly compare MNP’s efficacy with other antivirals (e.g., Paxlovid) used in similar patient cohorts. This would contextualize the findings and highlight MNP’s relative advantages or limitations.
Author Response
Please find attached a point-by-point response to referee's comments
Thank you very much

Reviewer 2 Report
Comments and Suggestions for Authors
Gentile et al. conducted an insightful study on the efficacy of molnupiravir in reducing the risk of severe outcomes in patients with SARS-CoV-2 infection. However, before the manuscript can be accepted, the following points should be addressed:
- L16 and throughout the manuscript: "Molnupiravir" should be written in lowercase as “molnupiravir.”
- Introduction section: There is limited information on molnupiravir, especially regarding its mechanism of action. Additional supporting information should be included, and you can refer to this article for more details: https://doi.org/10.3390/ijms25020739.
- Study design and population: Ethical approval for this study is not mentioned—please provide this information.
- L54: The inclusion criteria are mentioned, but what about the exclusion criteria?
- L68: Could you clarify which demographic data were collected for the study?
- L72: What is the Monoclonal Antibody Screening Score (MASS), and why was it used? There is no mention of this score in the results and discussion sections.
- L85-86: How was the composite outcome evaluated? Is there any reference to support this approach?
- L171-174: Please discuss the differences between your findings and those of Yip et al. and Najjar-Debbiny et al.
- L246: How might recall bias from telephone interviews have influenced the study findings?
Comments on the Quality of English Language
The English is fine and does not require any improvement.
Author Response

(The authors gave the same response as above.)

Reviewer 3 Report
Comments and Suggestions for Authors
Ivan Gentile et al. reported an interesting retrospective study. The paper discussed the outcome of SARS-CoV-2 management, and the topic fell within the scope of Microorganisms. The manuscript deserved publication after a Minor Revision. Please refer to the following comments.
- The significance of the findings should be highlighted at the end of Abstract.
- The Introduction read too short. More information should be introduced, in particular that of MNP. For example, what was the chemical structure of MNP? What were the advantages of MNP over other –ravir drugs?
- The p threshold value of significant different must be stated in Section 2.4.
- According to Table 1 and the related text, the age and health conditions of MNP and no therapy group varied a lot. Would this exert severe impact on the results? Please make proper comment on this issue.
- Please double-check whether the Standard Error 6.2 was correct in Table 3.
Besides, the authors were suggested to consult the Editorial Office whether the current style of Acknowledgement was acceptable.
Author Response

(The authors gave the same response as above.)

Round 2
Reviewer 1 Report
Comments and Suggestions for Authors
The main observations highlighted were considered by the authors, therefore we consider that the manuscript can be accepted for publication.
Reviewer 2 Report
Comments and Suggestions for Authors
Based on the reviewers' feedback, I believe the article is now ready for publication.